# Synthetic and genomic regulatory elements reveal aspects of *cis*-regulatory grammar in mouse embryonic stem cells

Dana M King[1,2†], Clarice Kit Yee Hong[1,2], James L Shepherdson[1,2], David M Granas[1,2], Brett B Maricque[1,2‡], Barak A Cohen[1,2]*

[1]Edison Center for Genome Sciences and Systems Biology, Washington University in St. Louis, St. Louis, United States; [2]Department of Genetics, Washington University in St. Louis, St. Louis, United States

*For correspondence:
cohen@wustl.edu

Present address: †University of Michigan, Bioinformatics Core, Ann Arbor, United States; ‡Columbia University, Department of Psychology, New York, United States

Competing interests: The authors declare that no competing interests exist.

**Abstract** In embryonic stem cells (ESCs), a core transcription factor (TF) network establishes the gene expression program necessary for pluripotency. To address how interactions between four key TFs contribute to *cis*-regulation in mouse ESCs, we assayed two massively parallel reporter assay (MPRA) libraries composed of binding sites for SOX2, POU5F1 (OCT4), KLF4, and ESRRB. Comparisons between synthetic *cis*-regulatory elements and genomic sequences with comparable binding site configurations revealed some aspects of a regulatory grammar. The expression of synthetic elements is influenced by both the number and arrangement of binding sites. This grammar plays only a small role for genomic sequences, as the relative activities of genomic sequences are best explained by the predicted occupancy of binding sites, regardless of binding site identity and positioning. Our results suggest that the effects of transcription factor binding sites (TFBS) are influenced by the order and orientation of sites, but that in the genome the overall occupancy of TFs is the primary determinant of activity.

## Introduction

### Independence versus interaction of transcription factor binding sites

Enhancers are composed of combinations of transcription factor binding sites (TFBS). An important question is: to what extent do TFBS act independently within enhancers and to what extent do specific interactions between transcription factors (TF) underlie enhancer function? Independence suggests a modular genome in which the effects of multiple binding sites are predictable from their individual effects. Interactions, such as cooperativity between TFs, cause the effect of multiple TFBS to be more (or less) than the combination of their individual effects. Constructing models that predict the expression of genes based on the TFBS composition of their surrounding regulatory DNA will require understanding the degree to which sites function independently and how interactions between sites contribute to the activity of regulatory sequences.

### Regulatory grammar

The extent to which TFs function either independently or through interactions should be reflected in the *cis*-regulatory *grammar* of TFBS, defined as the ways that the order, orientation, spacing, and affinity of binding sites impact the activity of enhancers. If TFs function independently then we do not expect strong constraints on the positioning of their binding sites within regulatory elements. If TFs function mostly through interactions with other TFs that require a precise geometry, then we expect strong biases in the positioning of TFBS within regulatory elements. At least three models make predictions of how grammar might influence enhancer activity, the billboard model, the

**eLife digest** Transcription factors are proteins that flip genetic switches; their role is to control when and where genes are active. They do this by binding to short stretches of DNA called *cis*-regulatory sequences. Each sequence can have several binding sites for different transcription factors, but it is largely unclear whether the transcription factors binding to the same regulatory sequence actually work together.

It is possible that each transcription factor may work independently and there only needs to be critical mass of transcription factors bound to throw the genetic switch. If this is the case, the most important features of a *cis*-regulatory sequence should be the number of binding sites it contains, and how tightly the transcription factors bind to those sites. The more transcription factors and the more strongly they bind, the more active the gene should be. An alternative option is that certain transcription factors may work better together, enhancing each other's effects such that the total effect is more than the sum of its parts. If this is true, the order, orientation and spacing of the binding sites within a sequence should matter more than the number.

One way to investigate to distinguish between these possibilities is to study mouse embryonic stem cells, which have a core set of four transcription factors. Looking directly at a real genome, however, can be confusing and it is difficult to measure the effects of different *cis*-regulatory sequences because genes differ in so many other ways. To tackle this problem, King et al. created a synthetic set of *cis*-regulatory sequences based on the four core transcription factors found in mouse stem cells.

The synthetic set had every combination of two, three or four of the binding sites, with each site either facing forwards or backwards along the DNA strand. King et al. attached each of the synthetic cis-regulatory sequences to a reporter gene to find out how well each sequence performed. This revealed that the cis-regulatory sequences with the most binding sites and the tightest binding affinities work best, suggesting that transcription factors mainly work independently.

There was evidence of some interaction between some transcription factors, because, of the synthetic sequences with four binding sites, some worked better than others, and there were patterns in the most effective binding site combinations. However, these effects were small and when King et al. went on to test sequences from the real mouse genome, the most important factor by far was the number of binding sites.

Synthetic libraries of DNA sequences allow researchers to examine gene regulation more clearly than is possible in real genomes. Yet this approach does have its limitations and it is impossible to capture every type of *cis*-regulatory sequence in one library. The next step to extend this work is to combine the two approaches, taking sequences from the real genome and manipulating them one by one. This could help to unravel the rules that govern how *cis*-regulatory sequences work in real cells.

enhanceosome model, and the TF collective model (*Kulkarni and Arnosti, 2003*; *Spitz and Furlong, 2012*). The enhanceosome model posits extensive interactions between bound TFs, resulting in a strict grammar in which only precise positioning of TFBS activate target genes. The enhanceosome model is supported by structural studies of the IFN-β enhancer, where a specific order and spacing of TFBS is required to activate expression (*Panne, 2008*; *Yie et al., 1999*). In contrast, the billboard model posits a more flexible grammar, where enhancers tolerate changes to the order, spacing, or orientations of TFBS with little change to target gene expression (*Giorgetti et al., 2010*; *Kulkarni and Arnosti, 2003*). In the billboard model bound TFs function in a largely independent manner. This model was proposed to explain binding site turnover in developmental enhancers and functional conservation of enhancer activity between species despite sequence divergence (*Hare et al., 2008a*; *Hare et al., 2008b*; *Ludwig et al., 2000*; *Visel et al., 2009*). In the TF collective model, specific TFs must be recruited to enhancers but can be recruited either by direct contact with DNA or indirectly through other TFs (*Junion et al., 2012*; *Spitz and Furlong, 2012*; *Uhl et al., 2016*). In the collective model no specific TFBS is required for activity even though the recruitment of individual TFs might be. TFs may function independently in some contexts and may engage in interactions in other contexts. The billboard, enhanceosome, and collective models differ in the

importance the precise arrangements of TFBS play in setting the activities of enhancers, and control of gene expression likely incorporates aspects of all three models. Quantifying the extent to which grammar influences activity in different contexts is an important step toward producing more predictive models of gene expression.

We and others have used mouse embryonic stem cells (mESCs) as a system for studying *cis*-regulatory grammar and cooperative interactions between the pluripotency factors POU5F1 (OCT4), SOX2, ESRRB, and KLF4 (*Dunn et al., 2014*; *Fiore and Cohen, 2016*; *Williams et al., 2004*). The pluripotency factors are a core set of TFs that maintain pluripotency in mESCs and are sufficient to induce pluripotency in terminally differentiated cells (*Feng et al., 2009*; *Liu et al., 2008*; *Niwa, 2014*; *Takahashi and Yamanaka, 2006*; *Zhang et al., 2008*). The pluripotency TFs activate self-renewal genes and repress genes that promote differentiation (*Chambers and Tomlinson, 2009*). Based on known physical and genetic interactions, as well as genome-wide binding assays, multiple interacting TFs specify target gene expression in mESCs (*Huang et al., 2009*; *Niwa, 2014*; *Reményi et al., 2004*; *Reményi et al., 2003*; *Williams et al., 2004*). However, it remains unclear how pluripotency TFs collaborate to drive-specific patterns of gene expression in ESCs, and what role, if any, is played by TFBS grammar in determining specificity in the genome (*Chambers and Tomlinson, 2009*; *Chen et al., 2008b*). Understanding how these factors combine to regulate their target genes is central to understanding the establishment and maintenance of the pluripotent state.

We previously addressed these questions by assaying a set of synthetic *cis*-regulatory elements that represent a small fraction of the possible arrangements of pluripotency TFBS. We identified some evidence for a grammar that is constrained by TFBS arrangement, including OCT4-SOX2 interactions. However, our previous study lacked sufficient power to detect other interactions (*Fiore and Cohen, 2016*). Here, we explore the role of grammar for pluripotency TFBS by assaying an exhaustive set of synthetic *cis*-regulatory elements, composed of TFBS for SOX2, OCT4, KLF4 and ESRRB, as well as a limited set of genomic regulatory sequences with comparable configurations of binding sites. The pattern of expression of synthetic regulatory elements is well predicted by a model that incorporates binding site position. However, despite all genomic sequences overlapping ChIP-seq peaks for at least one of the four pluripotency factors, only about a third of sequences drove reporter gene activity above background levels. Additionally, the positional grammar learned from synthetic sequences performed poorly in predicting the activity of genomic sequences. Genomic sequences appear to also include sequence features that recruit additional TFs, either directly through TF-DNA interactions or possibly indirectly through TF-TF interactions. Our results suggest that in the genome the overall occupancy of TFs is the best predictor of binding site activity. Our results with synthetic elements suggest that other aspects of grammar (order, orientation) can tune the activity of sites, but these effects are difficult to observe without direct experimental manipulations. In the genome only the number and affinity of sites shows a correlation with activity.

## Results

### Rationale and description of enhancer libraries

We designed two reporter gene libraries to explore the role of grammar in regulatory elements controlled by the pluripotency TFs. The first library, synthetic (SYN), contains a set of synthetic combinations of consensus TFBS for OCT4 (O), SOX2 (S), KLF4 (K), and ESRRB (E). We did not include sites for NANOG in our libraries as its position weight matrix (PWM) has low information content and is not amenable to a synthetic binding site approach. Nanog also appears to be dispensable for reprogramming terminal cells to a pluripotent state (*Wang et al., 2013*; *Wang et al., 2012*; *Jauch et al., 2008*; *Pan and Thomson, 2007*; *Takahashi and Yamanaka, 2006*). We did not incorporate MYC-binding sites in our libraries because MYC often acts independently of the core pluripotency TFs (*Chen et al., 2012*; *Chen et al., 2008c*; *Liu et al., 2008*).

We designed the SYN library to test how interactions between different TFs (heterotypic interactions) determine the activities of regulatory elements. If heterotypic interactions depend on the geometry of TF binding, then the order, orientation, and spacing of sites should influence activity. To test this prediction, we designed the SYN library to assay different orders and orientations of the pluripotency binding sites. The SYN library includes all possible 624 unique combinations of two, three, and four TFBS (2-mers, 3-mers, and 4-mers, respectively), with each TFBS in either the forward

or reverse direction (*Supplementary file 1A*). Each synthetic element in the SYN library contains no more than one copy of a given TFBS. We chose this library design to focus on heterotypic interactions and to avoid the confounding effects of homotypic interactions, which we examined in detail in a previous study (*Fiore and Cohen, 2016*). We embedded each TFBS in a constant 20 bp sequence with fixed spacing between sites to ensure that all the sites sit on the same side of the DNA helix. We avoided varying the length of the spacer sequence between sites because increasing the length of spacer sequences risks introducing cryptic binding sites that confound the results. For each TF, we used a consensus binding site based on its position weight matrix (PWM) in the JASPAR database (*Sandelin, 2004*; *Fiore and Cohen, 2016*). We did not vary the predicted affinity of the sites in the SYN library because we could not assay a library large enough to vary the affinity of sites while still testing all possible arrangements of sites. Our rationale was to retain the maximum power to detect the effects of the order and orientation of sites, and this required us to compromise on our ability to detect the effects of the spacing and affinity of sites. The highly controlled nature of the SYN library provides maximum power to detect interactions mediated by the order and orientation of sites.

The second library includes sequences from the mouse genome that match, as best as possible, members of the SYN library. Using the same PWMs used to design the SYN library, we scanned the mouse genome for combinations of the TFBS for O, S, K, and E within 100 bp of regions bound by any of the four pluripotency TFs in E14 mESCs as measured by ChIP-seq (*Fiore and Cohen, 2016*; *Bailey et al., 2009*; *Chen et al., 2008c*). We chose genomic sequences that contain one and only one binding site that scores above the PWM threshold for each factor to mimic the composition of the SYN library. We identified few clusters that included all four binding sites (<70). We therefore selected 407 genomic sequences with three pluripotency TFBS that could be compared to the exhaustive set of synthetic 3-mer elements. The resulting genomic wild-type library (gWT) is composed of 407 unique genomic sequences with combinations of any three of the four TFBS, with each site represented no more than once per sequence (Materials and methods, *Supplementary file 1E-F*). Although these sequences differ from SYN elements in the individual site affinities, spacings between TFBS, as well as intervening sequence composition, our expectation was that the gWT sequences would test how well interactions learned from the SYN library apply to genomic sequences. To confirm that the activity of the gWT sequences depends on the presence of pluripotency TFBS, we generated matched genomic mutant sequences (gMUT) in which all three of the identified pluripotency TFBS were mutated by changing two positions in each TFBS from the highest information content base to the lowest information base according to the PWM (*Figure 1—figure supplement 1*). The final gMUT sequences lack detectable TFBS for O, S, K, or E when rescanned with the threshold used to select the gWT sequences. The combined gWT/gMUT library allows us to quantify the contributions of the pluripotency sites to regulatory activity, as well as sample configurations of pluripotency TFBS from the genome that may provide insight into grammar for these sequences.

## MPRA of reporter gene libraries

We assayed the *cis*-regulatory activity of the SYN and gWT/gMUT libraries in mESCs using a plasmid-based Massively Parallel Reporter Assay (MPRA) (*Kwasnieski et al., 2012*). Each unique library member described above is present eight times with a different unique sequence barcode (BC) in its 3' UTR (*Fiore and Cohen, 2016*). The elements were placed directly upstream of a minimal promoter, mirroring classical tests of enhancer activity. The assay does not, however, test whether elements can function as long-range enhancers. To determine the relative activity of each sequence compared to the minimal promoter included in each construct, we included copies of plasmids with only the minimal promoter paired with over a hundred unique BCs in each library (Materials and methods). Our measurements were highly reproducible between biological replicates, with $R^2$ between 0.98 and 0.99 for replicates of the SYN library and 0.96–0.98 for the gWT/gMUT library, and are not driven by abundance biases in the library (*Figure 1—figure supplement 2*). After thresholding on DNA and RNA counts, we recovered reads for 100% (624/624) of our SYN elements and 99% (403/407) of paired gWT/gMUT sequences. The high concordance between replicates and simultaneous sequencing of the two libraries allowed us to make quantitative comparisons, both within and between libraries.

## Synthetic and genomic libraries support different grammar models

TFBS in synthetic regulatory elements make strong independent contributions to expression. Most synthetic elements drive expression over basal activity regardless of the number, order, or orientation of sites within the element (*Figure 1A*). Of all SYN elements, 77% (6% of 2-mers, 66% of 3-mers, 92% of 4-mers) were statistically different from basal levels in all three replicates after correcting for multiple hypothesis testing (Wilcoxon rank-sum test; Bonferroni correction, n = 637; p-values reported in *Supplementary file 1C*). In most cases, three or four consensus binding sites are sufficient to increase expression above basal levels, which suggests strong independent contributions of TFBS to the activity of synthetic elements. Synthetic elements with more binding sites generally drive higher expression than elements with fewer binding sites, supporting the idea that TFBS can contribute to expression in an independent and additive manner. However, the wide range of expression levels observed from different 4-mer elements must be due to the arrangement of the TFBS, as site number, identity, and affinity are fixed. The strong positive effect of adding sites demonstrates an independent effect of TFBS, while the diversity of expression among elements with the same number of sites reveals that grammar can quantitatively modulate activity.

In contrast to the synthetic elements, most genomic sequences in the gWT library did not exhibit regulatory activity above basal levels. Only 28% (113/403) of wild type genomic sequences were statistically different from basal levels in all three replicates (p<0.05, Wilcoxon rank-sum test; Bonferroni correction, n = 403; p-values reported in *Supplementary file 1H*). This low fraction of active gWT sequences is consistent with observations from functional tests of genomic sequences bound by key

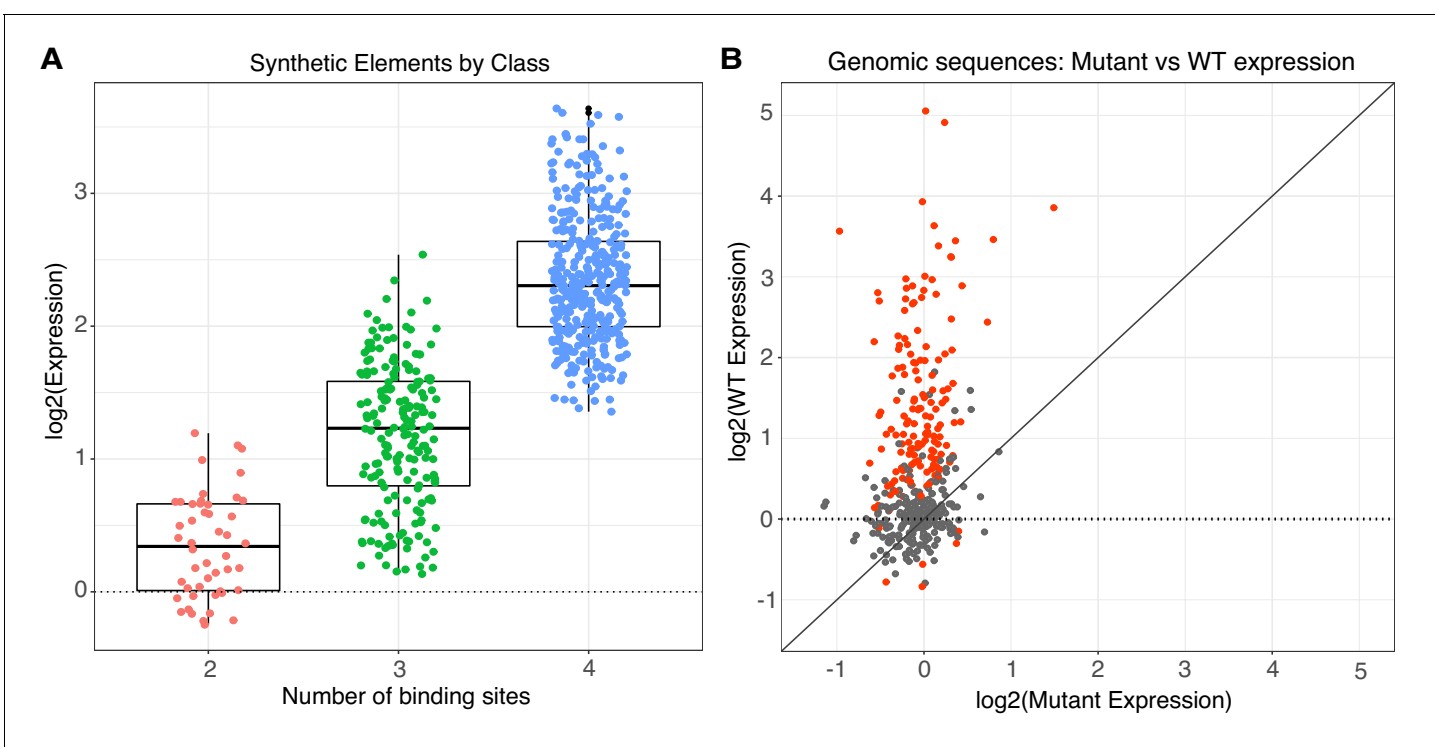

**Figure 1.** Activity of synthetic elements and genomic sequences. (**A**) The activity of synthetic elements with different numbers of binding sites. Expression is the average log of the ratio of cDNA barcode counts/DNA barcode counts for each synthetic element normalized to basal expression (dotted line). (**B**) The activity of genomic sequences is largely dependent on the presence of pluripotency binding sites. Normalized expression of wild type (gWT) sequences is plotted against expression of matched sequences with all three pluripotency TFBS mutated (gMUT sequences). Red indicates sequences with significantly different expression between matched gWT and gMUT sequences. The diagonal solid line is the expectation if mutation of TFBS had no impact on expression level. Expression of both gWT and gMUT sequences are normalized to basal controls, but basal expression is only plotted for gWT sequences on the y-axis (dotted line).

The online version of this article includes the following figure supplement(s) for figure 1:

**Figure supplement 1.** Pluripotency motif substitutions for gMUT sequences.
**Figure supplement 2.** MPRA data quality.

TFs in other cell types (*Fisher et al., 2012*; *Grossman et al., 2017*; *White et al., 2013*). The difference between the SYN and gWT libraries is that the surrounding sequence context in which the pluripotency sites occur in the gWT library varies much more than in the SYN library, and these contextual differences appear to have strong effects on the pluripotency sites. In most cases, the effect of sequence context in the gWT library was strong enough to suppress the independent contributions of the binding sites to activity. For genomic sequences that were statistically different from basal, 99% (112/113) have a significant difference between matched gWT and gMUT sequences (*Figure 1B*; p<0.05, Wilcoxon rank-sum test; Bonferroni correction, n = 403; p-values reported in *Supplementary file 1H*), indicating that the activity of these genomic sequences depends on one or more of the pluripotency TFBS. Our observation that the presence of high-quality pluripotency TFBS is generally insufficient to drive expression demonstrates that binding sites must be presented in the proper surrounding sequence context in order to generate a functional regulatory element.

## Synthetic elements support a positional grammar

While the overall pattern of expression of SYN elements supports strong independent contributions from binding sites, direct comparisons of different TFBS configurations also support a role for interactions between factors. Pairwise comparisons between 3-mers and their matched 4-mers that include one additional site at either the 5′ or 3′ end, reveal that the position of the extra site can strongly influence expression. For example, the O-K-E 3-mer and the matched O-K-E-S 4-mer drive indistinguishable expression, while the matched S-O-K-E 4-mer drives one of the highest expression levels in the SYN library (*Figure 2A*). Other examples are consistent with either strong position dependence or both position and orientation dependence (*Figure 2—figure supplement 1A–B*). Taken together, these results show that when an additional TFBS is added to an existing synthetic element, the position and orientation of the new site can have large effects on activity.

Synthetic elements appear to follow a grammar that includes some position specific interactions between TFBS. The ten highest expressing elements in the SYN library all have S and O sites next to each other and in the first two positions (*Figure 2B*), while the ten lowest expressing 4-mers have a strong bias for O and S in the last two positions (*Figure 2C*). The 10 highest expressing 4-mers all have K followed by E in the last two positions, while the lowest expressing 4-mers tend to have K and E in the first two positions. The fourth position can have an especially large effect on expression. In the highest 25% of 4-mers S is depleted (0/96) in the fourth position (*Figure 2D*), while in the lowest 25% E is virtually depleted (1/96) in the fourth position (*Figure 2E*). Conversely, in the fourth position, E is overrepresented in the top 25% (64/96) while S is overrepresented in the bottom 25% (48/96). These patterns also hold for comparisons of the strongest and weakest 3-mer and 2-mer elements (*Figure 2—figure supplement 1C–F*). These patterns indicate a grammar that includes a bias for S and O sites positioned upstream of K and E sites. This positioning may favor interactions between these factors and the basal transcriptional machinery or TFs recruited by the minimal promoter. As specifying a site at a given position restricts possible sites in neighboring positions, these patterns could also represent favorable interactions between factors. These data show that the precise arrangement of TFBS influences the activities of synthetic elements.

## Modeling supports a role for TFBS positions in setting expression level for synthetic elements but not for genomic sequences

While the grammar of O, S, K, and E sites influences the relative activities of the SYN elements, their order and orientation does not appear to contribute to the activity of genomic sequences. We compared the SYN and gWT libraries for elements with configurations of OKE, OSE, OSK, and SKE TFBS. Unlike SYN 3-mer elements, all four classes of gWT sequences span the full range of expression levels observed for the entire library, with only OSK sequences having a higher average expression (*Figure 3—figure supplement 1A*). Thus, in genomic sequences, the same arrangement of sites embedded in different genomic contexts can either fail to drive detectable activity or drive expression higher than the highest SYN library member. To quantify the divergence in activities between genomic and synthetic elements directly, we matched gWT sequences with pluripotency TFBS-dependent activity to SYN elements with the corresponding order of TFBS. We observed no correlation in regulatory activity between matched site configurations, ($R^2$ = 0.001; *Figure 3—figure supplement 1B*). These data indicate that other variables contribute to the *cis*-regulatory activity of

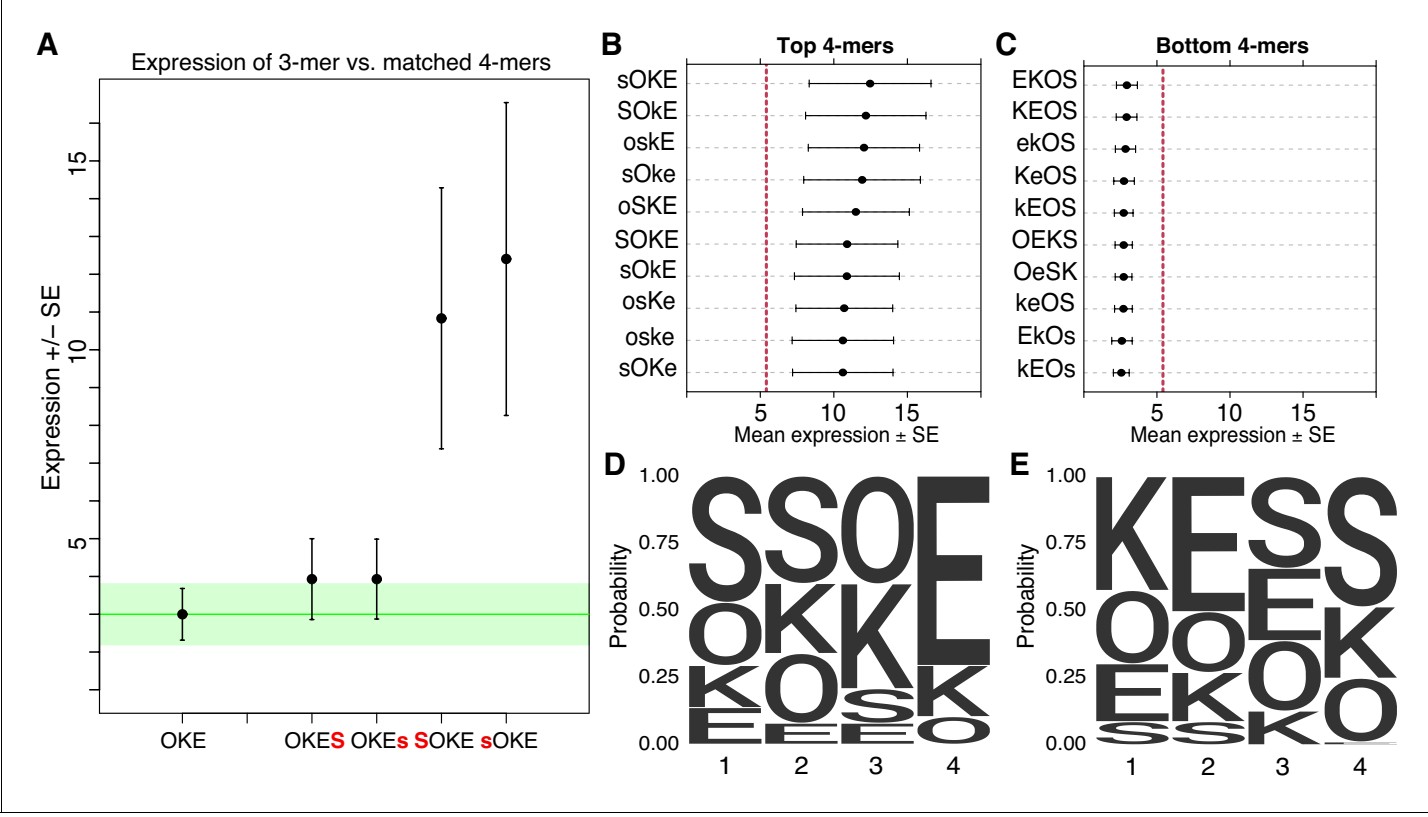

**Figure 2.** Non-additivity in synthetic elements. (**A**) Comparison of synthetic 3-mer elements with matched 4-mer elements containing one additional site in the first or fourth position. Mean expression of elements across barcodes (black dot) is plotted +/- SEM (black whiskers). Green line for comparison to expression of 3-mer; Green transparency highlights SEM of 3-mer shown. Capital letter represents binding site in forward orientation and lower-case letter represents binding site in reverse orientation. Activity of the ten highest (**B**) and ten lowest (**C**) expressing 4-mers. Red line represents average expression of all synthetic 4-mer elements. Case represents binding site orientation as in (**A**) Mean expression of each element across barcodes (black dot) +/- SEM (black whiskers). Activity logos for the top 25% (n = 96) (**D**) and bottom 25% (**E**) of 4-mer synthetic elements. Height of letter is proportional to frequency of site in indicated position. Positions organized from 5' end (Position 1) to 3' end (Position 4) of elements. The online version of this article includes the following figure supplement(s) for figure 2:

**Figure supplement 1.** Additional examples of non-additivity in synthetic elements.

gWT sequences, such as the spacing and affinities of the sites, or the presence of TFBS for additional factors in flanking sequences that are held constant in the SYN library.

To identify additional sequence features that might be contributing to activity, we used a variation of the Random Forest (RF) model, an unsupervised machine learning technique. RF models can be applied for either simple classification, assigning observations to group predictions, or classifying individual observations into semi-continuous bins to make quantitative, regression-case predictions. The accuracy of predictions are assessed over a large number of decision trees trained on random subsets of the data, which allows the contribution or 'variable importance' of specific features to be measured. As RFs are prone to biases from early random splits in the decision trees for unbalanced data, we used iterative Random Forests (iRF) as a tool for feature selection as well as for predicting activity (**Basu et al., 2018**).

We first trained a regression-case iRF model on the data from the SYN library. We initialized the models with four features (**Supplementary file 2A**), representing only the presence or absence of each of the four pluripotency TFBS. This 'independent' iRF model had an $R^2$ of 0.56 between observed and predicted observations when tested on held-out data for the final iRF iteration (**Figure 3—figure supplement 2**). However, the independent iRF model cannot account for the differences in activities between 4-mers, because all 4-mers have identical TFBS composition (4-mers $R^2$ = 0.00). To identify features that might distinguish between the activities of 4-mers, we trained an additional regression-case iRF model, 'independent + position', initialized with 20 features,

representing both the presence and position of the four TFBS in each SYN element (*Supplementary file 2A*). The 20-term positional model performs well in predicting SYN expression, with an overall $R^2$ of 0.87 for the last model iteration on a held-out test set (*Figure 3A*). The positional iRF model highly weights the presence/absence of the sites, as expected from the performance of the independent iRF model, but also has contributions from the presence of E in the 4th position and S in the first and second positions (*Figure 3B*). These results reinforce the conclusion that the activity of synthetic sequences depends both on the composition and positioning of TFBS.

iRF models trained on the SYN library failed to predict or classify the expression of genomic sequences. While synthetic elements had a range of activities, elements in the gWT library are predominantly inactive, and the small number of active gWT sequences drive expression across an order of magnitude of activity levels (*Figure 3—figure supplement 1A*). Having such a large number of inactive sequences in the pool makes it difficult to train a model that predicts the relative activities of genomic sequences. Retraining iRF regression models to predict gWT expression fails during the training step and has no correlation with the observed expression data (independent: $R^2 = 0.03$; independent + position: $R^2 = 0.001$). In all subsequent analyses of genomic sequences, we limited

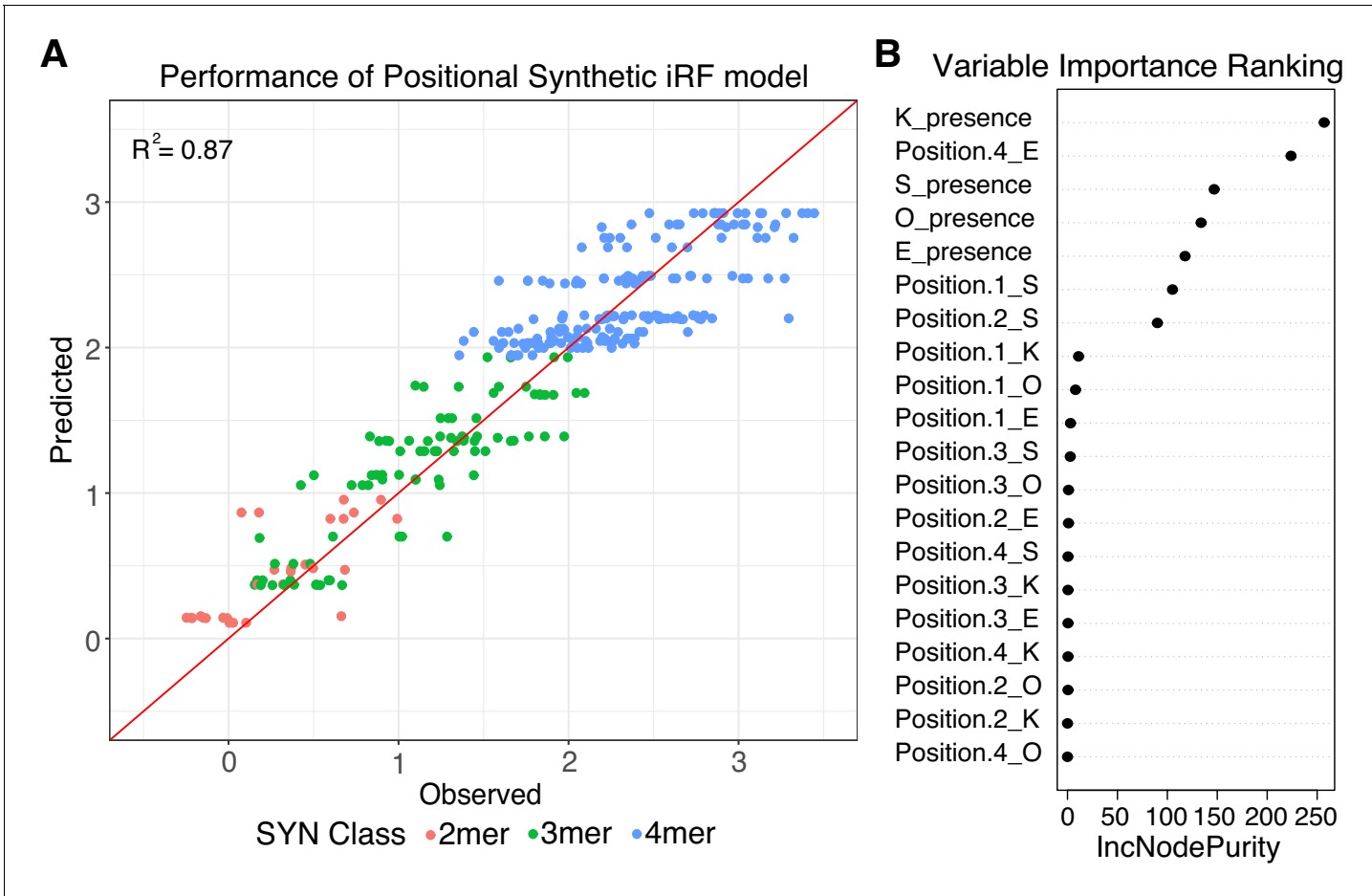

**Figure 3.** Positional grammar in synthetic elements. (**A**) Iterative random forest (iRF) regression model that includes features for presence and position of pluripotency TFBS predicts relative expression of synthetic elements. Number of binding site per element is indicated in pink (2-mers), green (3-mers), and blue (4-mers). Observed and predicted expression are both plotted in $\log_2$ space. (**B**) Ranking of variables in synthetic iRF model. Variable importance is estimated by Increased Node Purity (IncNodePurity), the decrease in node impurities from splitting on that variable, averaged over all trees during training.

The online version of this article includes the following figure supplement(s) for figure 3:

**Figure supplement 1.** Comparison of synthetic and genomic patterns of transcription factor binding sites (TFBS).

**Figure supplement 2.** Additive effects in synthetic elements.

**Figure supplement 3.** Effect of spacer sequences between TFBS on synthetic 4-mer expression.

ourselves to models that attempt to distinguish between active and inactive genomic sequences, without predicting the relative differences in activity among active sequences. However, our first attempt to produce a classifier failed. Training a classification model to distinguish between active and inactive gWT sequences (top 25%, n = 102; bottom 75%, n = 305) using either only independent or independent + position features also fails to perform better than chance (Independent: Area Under the Receiver Operator Curve (AUROC) = 0.52, Area Under the Precision Recall Curve (AUPRC) = 0.22; Positional: AUROC = 0.47, AUPRC = 0.25; *Supplementary file 2B*). Genomic and synthetic elements with the same pattern of sites can drive drastically different expression levels (*Figure 3—figure supplement 1B*). Other sequence features present in the flanking genomic sequences and absent from the synthetic elements must therefore play a role in setting activity levels, in addition to the identity and position of the individual pluripotency TFBS.

Our results with genomic elements suggested that the sequences flanking the pluripotency TFBS play a role in determining *cis*-regulatory activity. We tested the effect of changing spacer sequences that flank the TFBS in six 4-mer elements from the SYN library. We tested four different spacer sequences, for a total of 30 library members, which includes the original spacer sequence. The new spacers sequences were designed to match the nucleotide content of the original spacers and minimize the creation of new TFBS (*Supplementary file 1J*). To ensure the dynamic range of the library, we mixed this 'mini spacer library' library with a small portion of the SYN library and performed an MPRA.

We found that changing the spacer sequences in the SYN library had small, but significant effects on the activities of the 4-mers. The activities of all six 4-mers in the mini spacer library tested with all four spacer sequences remained in the original range of expression for 4-mers (*Figure 3—figure supplement 3A*). On average, the spacer sequences modified expression by 6% (0.3–25%, *Figure 3—figure supplement 3B*). Although the overall effects of spacer sequences were small, the rank order of the 4-mers did change for different spacers (*Figure 3—figure supplement 3C*), supporting the idea that sequence features flanking the binding sites do affect gene expression. These results are consistent with the differences between the SYN and gWT libraries.

## Site affinity contributes to the activity of genomic sequences

We attempted to identify other sequence features that might differentiate active and inactive gWT sequences. Sequence-based support vector machines (*k*mer-SVMs) are powerful tools to predict the activity of regulatory elements (*Fletez-Brant et al., 2013*; *Chaudhari and Cohen, 2018*). To identify sequence features that explain the differences between genomic elements, we trained a gapped *k*mer SVM (gkm-SVM) (*Ghandi et al., 2016*; *Ghandi et al., 2014*). The best performing gkm-SVM classified our positive and negative sets with AUROC of 0.75 and AUPRC of 0.77 (*k* = 8, gap = 2; *Figure 4A*). Although all sequences in the gWT library were selected to contain TFBS for the four pluripotency factors, many of the discriminative 8-mers (29/50) have motif matches that include at least one pluripotency family member (*Fletez-Brant et al., 2013*; *Bailey et al., 2009*; *Supplementary file 2D*). This suggests that the differences between active and inactive genomic sites could be due to the primary pluripotency sites or secondary occurrences of these sites in the intervening sequences that scored below the scanning threshold.

Sequences with higher predicted affinity pluripotency TFBS may drive higher expression. To determine if differences in the primary pluripotency sites are part of the signal identified by the SVM, we annotated gWT sequences with PWM-based scores for each TFBS present (*Grant et al., 2011*). For SOX2, we found no difference in scores between high and low sequences (*Figure 4B*; p=0.07, Welch's t-test). For OCT4, we found a modest difference between the average scores for high and low sequences and a broader but also a significant difference for KLF4 and ESRRB PWM scores (*Figure 4C–E*). Summing the PWM scores for all of the TFBS further separates high and low sequences (*Figure 4F–G*). These patterns suggest that the quality of the primary sites contributes to the activity differences observed among gWT sequences.

We then asked if secondary sites for the pluripotency TFs might contribute to *cis*-regulatory activity by calculating predicted occupancy for both gWT sequences and gMUT sequences that lack the primary binding sites (Materials and methods). Predicted occupancy is a metric that includes contributions from any primary, well-scoring TFBS plus contributions from weaker sites that might be missed with traditional motif scanning (*White et al., 2016*; *White et al., 2013*; *Evans et al., 2012*; *Segal et al., 2008*; *Zhao et al., 2009*). We found evidence for additional low predicted affinity sites

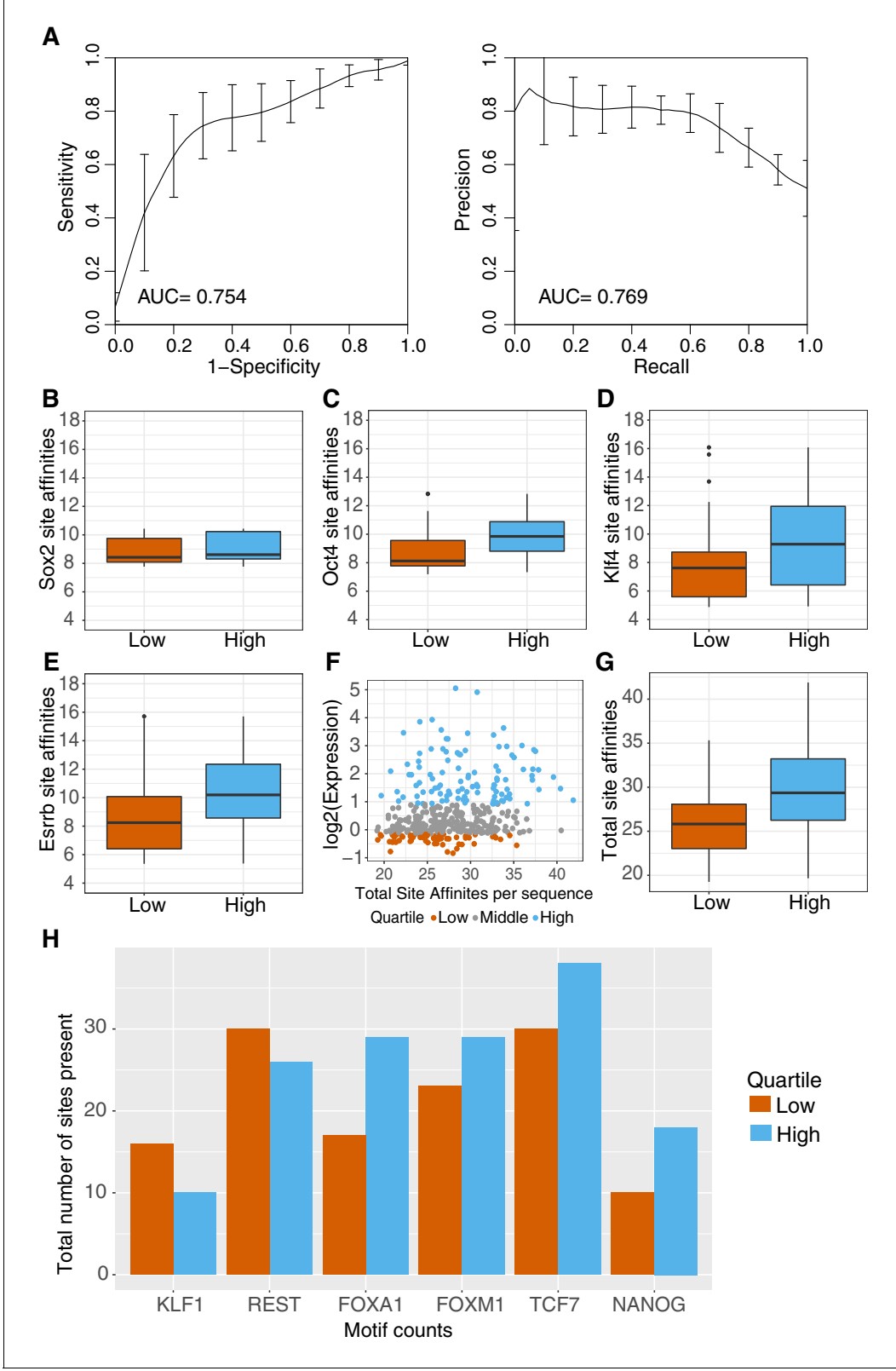

**Figure 4.** Sequence features separate active and inactive genomic sequences. (**A**) Performance of gkm-SVM for genomic sequences supports contribution of sequence-based features to activity. Word length of 8 bp with gap size of 2 bp was used for training with threefold cross validation. ROC curve (left panel) and PR curve (right panel) is plotted for the average across threefold cross-validation sets +/- standard deviation. (**B–E**) Primary (O,S,K,E) site affinities across gWT sequences, as output during motif scanning plotted for high genomic sequences (top 25% as ranked by expression,

*Figure 4 continued on next page*

*Figure 4 continued*

n = 101) and low genomic sequences (bottom 25% as ranked by expression, n = 101). (**F–G**) Total site affinities is calculated per sequence by summing the predicted affinity of the three primary sites present in each sequence. (**H**) Total number of occurrences of TFBS for additional TFs in high and low sequences (stratified as in **B–G**), as determined by motif scanning, excluding primary (O,S,K,E) sites.

The online version of this article includes the following figure supplement(s) for figure 4:

**Figure supplement 1.** Predicted occupancy of genomic sequences.

**Figure supplement 2.** Genomic sequences show distance preferences between factors.

for SOX2 and OCT4 in both high and low sequences, making it unlikely that low-affinity sites strongly contribute to expression differences (*Figure 4—figure supplement 1*). Together, these results suggest that the affinities of the primary sites in genomic sequences, which are fixed in synthetic elements, contribute to the regulatory activity of genomic sequences more than the presence of additional sites with low predicted affinity.

We also analyzed whether the spacing between binding sites correlated with the activity of *cis*-regulatory elements. Using the same annotations used to determine the predicted affinities of SOX2, OCT4, ESRRB, and KLF4 binding sites, we calculated the edge-to-edge distance between every possible pair of binding sites and plotted the frequency of each spacing for high and low activity sequences (*Figure 4—figure supplement 2*). We observed a preference in high activity sequences for closely spaced sites for OCT4 and SOX2 reflecting a known interaction between these TFs. We also observed preferences in high activity genomic sequences for closely spaced KLF4 and OCT4 sites, and for ESRRB and OCT4 sites. Binding site spacing may therefore play a role in setting the relative activities of genomic sequences.

## Contributions from sites for other transcription factors

A major difference between the synthetic and genomic elements is the presence of sites for TFs besides the pluripotency factors. While the synthetic elements were designed to keep the sequences between pluripotency sites constant, genomic sequences differ in both the length and composition of sequences between the pluripotency sites. The presence of binding sites for additional transcription factors may contribute to the activity of genomic sequences. To identify sites for other factors that could contribute to differences between high and low activity gWT sequences, we examined the top discriminative 8-mers from the gkm-SVM, looking at possible PWM matches for additional TFs (*Supplementary file 2D*). We then used PWMs for these additional TFs to identify instances of sites for other factors in the genomic sequences (see Materials and methods) (*Grant et al., 2011*; *Sandelin, 2004*). We found significant enrichment for FOXA1 sites (*Figure 4H*). We also found that FOXA1 and NANOG had higher total PWM scores in the high activity sequences (*Figure 5—figure supplement 1A*). While FOXA1 is likely not present in mESCs, other family members (FOXA2, FOXD1, FOXP1) are expressed in ESCs and have been shown to contribute to the pluripotent regulatory network, and therefore could be acting on the gWT sequences through these binding sites (*Pan and Thomson, 2007*; *Mulas et al., 2018*; *Gabut et al., 2011*).

Genomic sequences with higher occupancy by TFs in the genome, as measured by ChIP-seq, have higher average expression in our assay. We annotated the gWT intervals with publicly available ChIP-seq data for additional TFs and with ATAC-seq data from E14 mESCs to determine if differences in accessibility explained the difference between high and low activity sequences (*Supplementary file 2B*). Both high and low activity gWT sequences were accessible in the genome showing that accessibility does not necessarily correlate with high activity sequences. High activity sequences had a small but significant overlap with NANOG peaks (*Figure 5—figure supplement 1B*). However, for the 328 genomic sequences with a NANOG ChIP-seq signal, only 16% had an underlying TFBS as determined by motif scanning. Therefore, NANOG might be recruited by other pluripotency TFs to these sequences independent of high-quality TFBS for this factor. If we compare expression levels to the number of overlapping ChIP-seq peaks, including O,S,K,E and these additional TFs, we see that gWT sequences with higher occupancy in the genome have higher average expression in our assay (*Figure 5*), which has been previously observed in HepG2 cells (*Ulirsch et al., 2016*). This result supports a model where cumulative occupancy sets activity level.

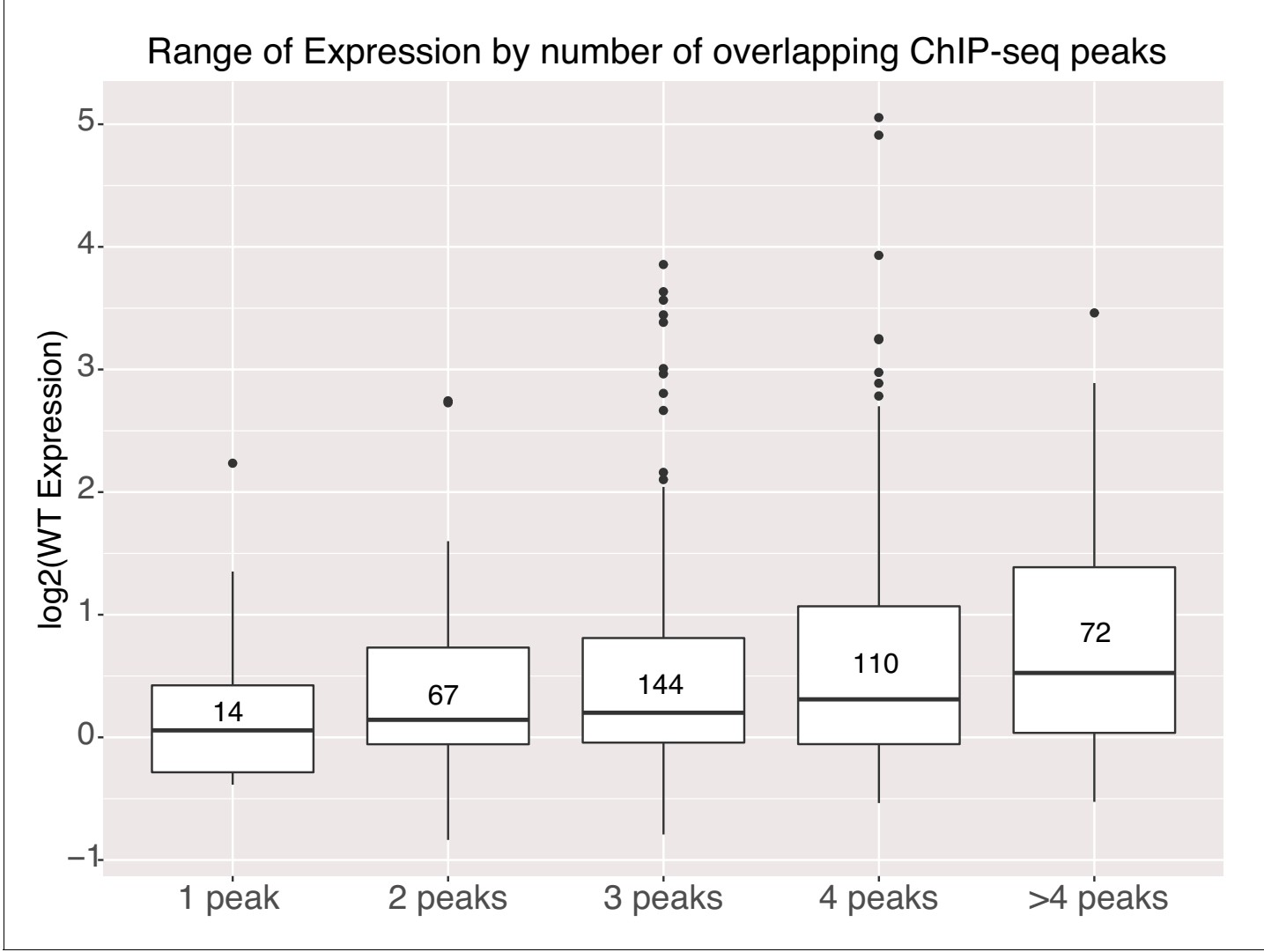

**Figure 5.** Activity of genomic sequences scales with increased occupancy in the genome. Expression of elements binned by number of intersected ChIP-seq peak signals for different factors. Number of sequences in each bin indicated in center of boxplot. All gWT sequences overlapped at least one ChIP-seq peak as per library design.

The online version of this article includes the following figure supplement(s) for figure 5:

**Figure supplement 1.** Genomic sequences show signatures for other factors.

To understand the relative contributions of the sequence features that were enriched individually, we trained iRF models with different subsets of these sequence features and compared their performance on a held-out test set (*Supplementary file 2B*). None of these models accurately predicted the activity of genomic sequences, likely because most genomic sequences in our collection had no activity above basal levels. Therefore, we attempted to classify active from inactive genomic sequences.

We trained an iRF model initialized with 58 features that capture differences between gWT sequences and SYN elements. These features include predicted affinity and preferred spacings between the pluripotency TFBS, the predicted occupancy for the pluripotency TFs, the presence of binding sites for additional TFs, plus chromatin accessibility (ATAC-seq) and ChIP-seq peaks for both TFs and histone marks, as well as summary features such as the total primary site affinities for each sequence (*Supplementary file 2B*). This gWT iRF model classified active from inactive on a held out test set with AUROC = 0.67, and AUPRC = 0.46 (*Figure 6A–B*, model 'All'). Models that only included subsets of features — the spacing between elements (model 'Spacing'), the strength of the

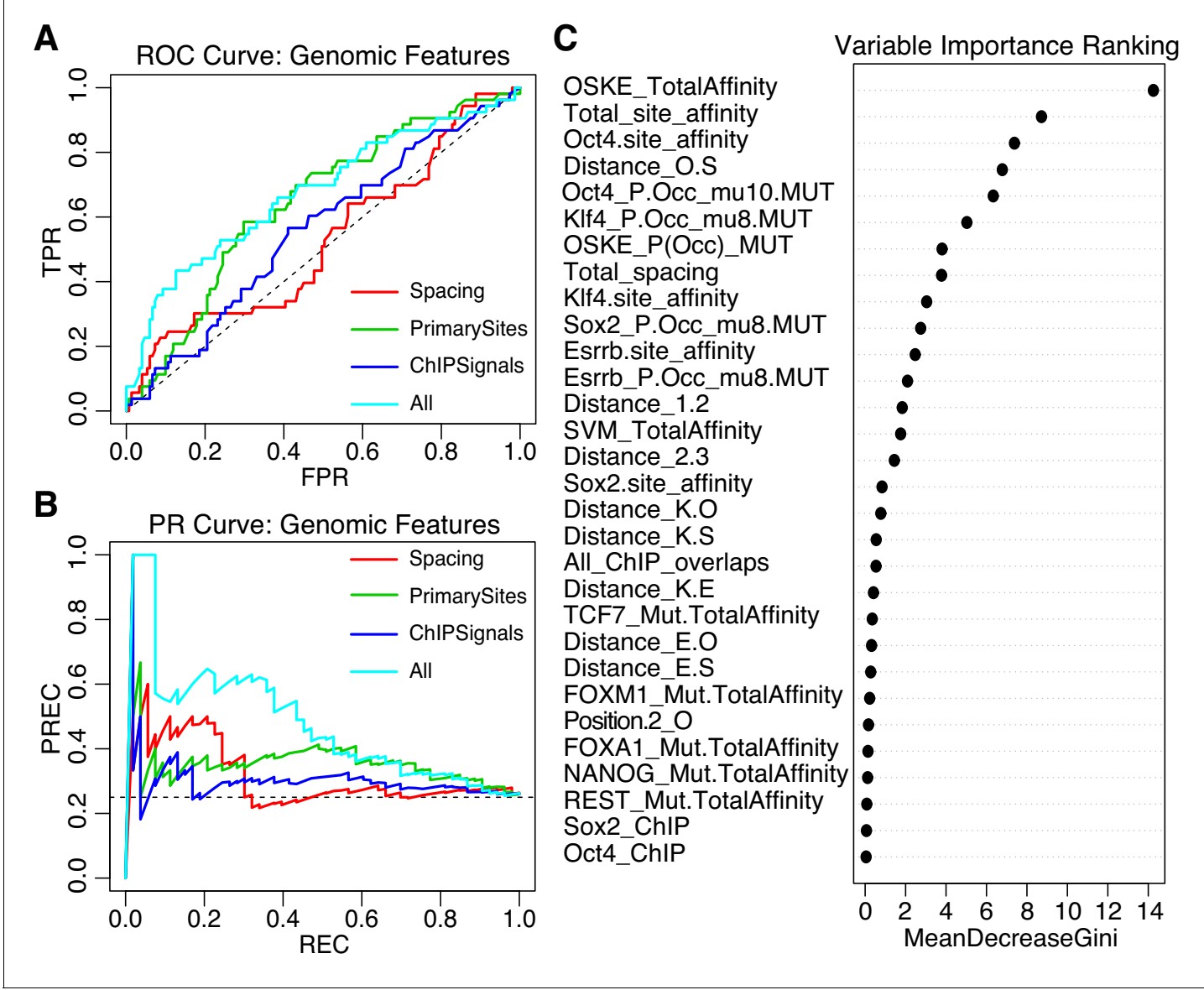

**Figure 6.** Performance of iRF classification models that include features specific to genomic sequences. (**A**) ROC Curve and (**B**) Precision-Recall (PR) Curve comparing genomic iRF models. Color indicates set of features used to train model. (**C**) Variable importance as evaluated for the feature by the average reduction in the Gini index (**Chen et al., 2008c**).

pluripotency sites ('PrimarySites'), or the overlapping ChIP signal ('ChIPSignals') — did not perform as well (**Figure 6A–B**). The features that best separate active from inactive sequences were related to attributes of the pluripotency sites with the top feature being the summed pluripotency factor predicted affinity per sequence ('OSKE_TotalAffinity', **Figure 6C**). Taken together, our data suggest that genomic sequences drive higher expression when they contain strong binding sites with pre-ferred spacing and are embedded in sequences that can mediate the recruitment of other TFs or cofactors.

## Discussion

In this study, we sought to understand how pluripotency factors collaborate to drive specific levels of expression by testing both an exhaustive set of synthetic arrangements of TFBS for OCT4, SOX2, KLF4, and ESRRB and comparable genomic sequences. The experimental design allowed for direct

comparisons between the regulatory grammar of synthetic and genomic sequences. The strongest similarity between synthetic and genomic elements is that in both cases activity depends heavily on the number and affinity of binding sites. These results are most consistent with a model in which the overall occupancy of a sequence by its cognate TFs is the primary determinant of that element's activity. Consistent with this hypothesis, the predictive power of our trained genomic model derived primarily from summing over the number and affinity of binding sites. We also observed correlation between the occupancy of sites as measured by ChIP-seq and their activity in MPRA assays. While there are many steps involved in activating gene expression, the occupancy model posits that the strength of a regulatory element is primarily controlled by its fractional occupancy by TFs.

The occupancy model might also explain the surprising result that the activity of genomic elements in our plasmid MPRA experiments do not correlate with experimental measurements of how accessible the chromatin is in their native locations. Plasmid assays might not capture regulation by chromatin, but in many cases plasmid assays do recapitulate the activity of chromosomally integrated elements (*Maricque et al., 2019*; *Inoue et al., 2017*). Alternatively, accessible regions may be bound by transcription factors but may not necessarily drive activity, such as in the case of 'poised' regulatory elements (*Cruz-Molina et al., 2017*). Nucleosome exclusion is important for regulatory activity (*Khoueiry et al., 2010*) and may reflect TF binding, but accessibility itself may not be sufficient for regulatory activity. Another possibility is that open chromatin may not be a direct reflection of the occupancy of an element by its cognate TFs. Other factors besides occupancy by TFs also determine the openness of chromatin, such as chromosome topology, the proximity of origins of replication, and nucleotide composition. This may explain why some genomic sequences with binding sites that reside in open chromatin do not drive high activity in MPRA assays. The prediction is that these regions are open for reasons other than occupancy by cognate TFs. That the activity of genomic elements correlates with TF occupancy as measured by ChIP-seq, but not necessarily open chromatin measurements by ATAC-seq, supports the occupancy model.

While TF occupancy was the best predictor of activity, the AUROC and AUPRC analyses show that we are still missing important features that underlie the activity of genomic sequences. Indeed, two-thirds of genomic sequences that contain consensus motifs and reside under a ChIP-seq peak for one of the pluripotency TFs had no activity in our assay. Why don't all sequences occupied by TFs have strong regulatory activity? The sequence context in which occupied binding sites occur must contribute heavily to their activity. We attempted to address this issue by examining the regulatory grammar of synthetic elements.

Synthetic elements provide a highly controlled system for exploring whether TFBS are constrained by a regulatory grammar. With synthetic elements we found clear evidence that their activity depends on the position and orientation of pluripotency binding sites. Synthetic elements with the same number and affinity of TFBS had different levels of activity depending on the order and orientation of the sites. This result suggests that active regulatory elements in the genome are defined not only by the presence of TF occupied motifs, but also by cues in the surrounding DNA sequences. However, our models that captured the specific regulatory grammar of synthetic elements failed to predict the activity of genomic sequences.

Why don't models that robustly predict the activity of synthetic elements also predict the activity of genomic sequences? With synthetic elements, each sequence differs from others in the library by only a small number of sequence features. In synthetic libraries, there are many pairs of elements that differ by only a single sequence feature, which provides power to observe experimentally the effect of a single variable. In contrast, libraries of genomic elements are much more diverse, and the analysis of genomic sequences relies on detecting correlations between elements that share sequence features. However, it is difficult to isolate the effect of a single sequence feature because genomic elements that share a certain sequence feature will always be very different in terms of other features. The strength of the synthetic approach is the power it provides to isolate the effects of specific sequence features or pairs of sequence features. The weakness of the synthetic approach is that genomic elements are subject to many context specific constraints, all of which cannot be captured in a single synthetic library. When we changed the spacer sequences in our synthetic library, we found small but reproducible effects on expression. Our interpretation of this result is that changing the spacer sequences did not have large effects on the independent contribution of each TFBS, but did have effects on the interactions between sites (i.e. the regulatory grammar). In the future, we plan to use the regulatory grammar derived from synthetic elements to design

experiments that manipulate single features of genomic elements. If the grammar that is learned from synthetic elements reflects real constraints in the cell, then models of synthetic elements should predict the relative effects of single perturbations of genomic elements even if they cannot predict the absolute expression of genomic sequences. A combined approach that leverages both synthetic and genomic sequences should continue to help unravel the rules that govern *cis*-regulation of expression in cells.

# Materials and methods

## Key resources table

| Reagent type (species) or resource | Designation | Source or reference | Identifiers | Additional information |
|---|---|---|---|---|
| Cell line (*Mus musculus* mouse) | RW4 | other | RRID:CVCL_6442 | Gift from Mitra Lab, CGS, Department of Genetics, Washington University School of Medicine in St. Louis. The cell line tested negative for mycoplasma contamination by the Genome Editing and iPSC core at Washington University in St. Louis. |
| Commercial assay or kit | PureLink RNA Mini Kit | ThermoFisher Scientific/Invitrogen | Cat#:12183018A | Followed manufacturer's protocol |
| Commercial assay or kit | PureLink DNase Set | ThermoFisher Scientific/Invitrogen | Cat#:12185010 | Followed manufacturer's protocol |
| Commercial assay or kit | TURBO DNA-free | ThermoFisher Scientific/Invitrogen | Cat#:AM1907 | Followed manufacturer's protocol |
| Commercial assay or kit | SuperScript III Reverse Transcriptase | ThermoFisher Scientific/Invitrogen | Cat#:18080044 | Followed manufacturer's protocol |
| Commercial assay or kit | anti-Alkaline Phosphatase (AP) staining | System Biosciences | Cat.#:AP100R-1 | Followed manufacturer's protocol |
| Recombinant DNA reagent | SYN | this paper | | Recombinant plasmid library of synthetic (SYN) elements upstream of a minimal Pou5f1 promoter and dsRed/SV40 UTR reporter element |
| Recombinant DNA reagent | GEN | this paper | | Recombinant plasmid library of sequences identified in the mouse genome (GEN) upstream of a minimal Pou5f1 promoter and dsRed/SV40 UTR reporter element |
| Recombinant DNA reagent | miniSpacer | this paper | | Recombinant plasmid library of synthetic elements with swapped spacer (miniSpacer) sequences upstream of a minimal Pou5f1 promoter and dsRed/SV40 UTR reporter element |
| Software, algorithm | Bedtools v2.2 | https://bedtools.readthedocs.io/en/latest/ | RRID:SCR_006646 | DOI: 10.1093/bioinformatics/btq033 |
| Software, algorithm | iRF v2.0.0 | https://cran.r-project.org/web/packages/iRF/index.html | DOI: 10.1073/pnas.1711236115 | |

*Continued on next page*

*Continued*

| Reagent type (species) or resource | Designation | Source or reference | Identifiers | Additional information |
|---|---|---|---|---|
| Software, algorithm | gkm-SVM | https://cran.r-project.org/web/packages/gkmSVM/index.html | DOI: 10.1093/bioinformatics/btw203 | |
| Software, algorithm | BEEML | http://stormo.wustl.edu/beeml/ | DOI: 10.1371/journal.pcbi.1000590 | |

## Library design

To generate a library that contained both synthetic and genomic elements, we ordered a custom pool of 13,000 unique 150 bp oligonucleotides (oligos) from Agilent Technologies (Santa Clara, CA) through a limited licensing agreement. Each oligo in the SYN pool was 150 bp in length with the following sequence:

CTTCTACTACTAGGGCCCA[SEQ]AAGCTT[FILL]GAATTCTCTAGAC[BC]TGAGCTCTACATGCTAGTTCATG

where [SEQ] is a 40–80 bp synthetic element comprised of concatenated 20 bp building blocks of pluripotency sites, as described previously, with the fifth position of the KLF4 site changed to 'T' to facilitate cloning (*Fiore and Cohen, 2016*). [FILL] is a random filler sequence of variable length to bring the total length of each sequence to 150 bp, and [BC] is a random 9 bp barcode. The oligonucleotide pool contained all possible combinations of the pluripotency binding sites in both orientations, with no more than one of each site per sequence in lengths of two, three, and four building blocks. The sequence of each of the element is listed in *Supplementary file 1B*. In total, the SYN library has 624 unique synthetic elements. Each synthetic element is present in the pool eight times, each time with a different unique BC. There are also 112 oligos in the pool for cloning the basal promoter without any upstream element, each with a unique BC.

Genomic sequences were represented in the pool by 150 bp oligos with the following sequences:

GACTTACATTAGGGCCCGT[SEQ]AAGCTT[FILL]GAATTCTCTAGAC[BC]TGAGCTCGGACTACGATACTG

where [SEQ] is either a reference (gWT) or mutated (gMUT) genomic sequence of 81–82 bps. Reference gWT sequences were selected by choosing regions of the genome within 100 bps of previously identified ChIP-seq peaks for these four pluripotency factors (*Chen et al., 2008b*). After excluding poorly sequenced and repetitive regions (*ENCODE Project Consortium, 2012*; *Waterston et al., 2002*), we scanned the remaining regions using FIMO with the four PWMs used previously to design the synthetic building blocks, with a p-value threshold of $1 \times 10^{-3}$ (*Grant et al., 2011*; *Bailey et al., 2009*; *Fiore and Cohen, 2016*). Regions that contained more than one overlapping site identified by FIMO were excluded. Binding sites that were located less than 20 bp from each other were then merged into a single genomic element using Bedtools (*Quinlan and Hall, 2010*). Elements with no more than one of each site per element were then selected and expanded to 81–82 bp centered on the motifs. Expanded sequences were rescanned to confirm the presence of only three binding sites with the same threshold as used to originally scan the sequences. Sequences that contained restriction sites for were then removed from the library, leaving 407 genomic sequences with combinations of the OCT4, SOX2, KLF4, and/or ESSRB TFBS.

We generated matched mutated sequences (gMUT) for each of the 407 gWT sequences by changing two positions in each motif from the highest information content base to the lowest information base for that position (*Figure 1—figure supplement 1*). The reverse complement position and substitution was made for the reverse orientation of each motif. The mutated sequences were rescanned with all four original PWMs to confirm that no detectable pluripotency TFBS remained, using FIMO with the same p-value threshold ($1 \times 10^{-3}$) as above.

In total, the pool of oligos representing genomic sequences contained 407 wild-type sequences (gWT) and the corresponding 407 gMUT sequences. The sequence of each element is listed in *Supplementary file 1G*. Each of these 814 sequences were associated with eight unique BCs. The primers for gWT and gMUT sequences were identical so all subsequent steps for this library was performed in a single pool. There are also 112 oligos in the pool for cloning the basal promoter without

any upstream element, each with a unique BC (*Supplementary file 1F*). The rest of the array contained sequences not used in this study.

## Cloning of plasmid libraries

For a full list of primers, see *Supplementary file 3*. The synthesized oligos were prepared as previously described (*Kwasnieski et al., 2012*; *Fiore and Cohen, 2016*), except using primers Synthetic_FW-1 and Synthetic_Rev-2 with an annealing temperature of 55°C for the SYN library and primers Genomic_FW-1 and Genomic_Rev-1 with an annealing temperature of 53°C for the gWT/gMUT libraries. PCR products were purified from a polyacrylamide gel as described previously (*White et al., 2013*). Each library was cloned as described previously (*Fiore and Cohen, 2016*), with an SYN element (SYN library) or either a gWT or gMUT sequence (gWT/gMUT library) cloned into the ApaI and SacI sites of plasmid pCF10.

The *pou5f1* basal promoter and dsRed reporter gene were amplified from pCF10 using primers CF121 and CF122, and inserted into the plasmid library pools from the previous step at the XbaI and HindIII sites. Digestion of the libraries with SpeI and subsequent size selection was omitted as the SYN library had less than 2% background and the combined gWT/gMUT library had less than 1% background in the final cloning step.

## Spacer library

For the mini spacer library, we ordered an oligo pool containing 4-mer elements with different spacer sequences from Integrated DNA Technologies (Coralville, IA). Each oligo in the mini library was 161 bp in length with the following sequence:

GACATCAAGATCTGGCCTCGGGGCCC[SEQ]AAGCTTGAATTCTCTAGAC[BC]TGAGCTCTCGC
TTCGAGCAGACATGAT

where [SEQ] represents an oligo sequence described below and [BC] is a random 9 bp barcode. We picked six 4-mer oligos from the original synthetic library to span the 4-mer expression range and swapped out the spacer sequences in the oligos for four other sequences, generating a total of 30 constructs, including the original spacers. Each construct was represented in the pool with five unique barcodes. The sequence of each element is in *Supplementary file 1K*.

The mini spacer library was cloned into the same backbone as the previous libraries. Briefly, pCF10 was digested with ApaI and SacI, and the single-stranded oligo pool was directly assembled into the backbone using HiFi DNA assembly The *pou5f1* basal promoter and dsRed reporter gene were amplified from pCF10 using CF121 and CF122, then ligated into the mini spacer library following the same approach as the SYN, gWT, and gMUT libraries.

## Cell culture and transfection

RW4 mESCs were cultured as described previously (*Xian et al., 2005*; *Chen et al., 2008a*) on 2% gelatin coated plates in standard media (DMEM, 10% fetal bovine serum, 10% newborn calf serum, nucleoside supplement, 1000 U/ml leukemia inhibitory factor (LIF), and 0.1 μM B-mercaptoethanol). Approximately 1 million cells at 100% estimated viability were seeded into six-well plates 24 hr prior to transfection. The SYN library and combined gWT/gMUT were transfected in parallel using 10 μL Lipofectamine 2000 (Life Technologies, Carlsbad, CA), 3 μg of plasmid library, and 0.3 μg CF128 (a GFP control plasmid) per well, as described previously (*Fiore and Cohen, 2016*). Four biological replicates of each library pool, the SYN plasmid pool or combined gWT/gMUT plasmid pool, were transfected and the plates were passaged 6 hr post-transfection. For three replicates of each library pool, RNA was extracted 24 hr post-transfection from approximately 9 million cells per replicate, using the PureLink RNA mini kit (Life Technologies, Carlsbad, CA) with the fourth transfection replicate reserved for estimating transfection efficiency via fluorescent microscopy and staining for alkaline phosphatase (AP) activity, a universal pluripotency marker (*Singh et al., 2012*).

## Massively parallel reporter assay

Massively parallel reporter gene assays were used to measure the activity of each element as described previously (*Fiore and Cohen, 2016*; *Mogno et al., 2013*). Briefly, we used Illumina Next-Seq (San Deigo, CA) sequencing of both the RNA and original plasmid DNA pool, removing excess

DNA from the RNA pool using TURBO DNA-free kit (Life Technologies, Carlsbad, CA). cDNA was then prepared using SuperScript RT III (Life Technologies, Carlsbad, CA) with oligo dT primers. Both the cDNA and the plasmid DNA pool were amplified using primers CF150 and CF151b, for 13 cycles. The PCR amplification products were digested using XbaI and XhoI (New England Biolabs, Ipswich, MA), ligating the resulting digestion products to custom Illumina adapter sequences, P1_XbaI_X (where X is 1 through 8, with in-line multiplexing BC sequences) to the 5' overhang and PE2_SIC69_SalI on the 3' XhoI overhang, each of which is comprised of annealed forward (F) and reverse (R) strands. An enrichment PCR with primers CF52 and CF53 was then used, and the resulting products were mixed at equal concentration and sequenced on one NextSeq lane.

Sequencing reads were filtered to ensure that the BC sequence perfectly matched the expected sequence. For the SYN library, this resulted in 40 million reads combined for the three demultiplexed RNA samples (P1_XbaI_1, P1_XbaI_2, P1_XbaI_3; 12.7–13.5 million each), and 19.7 million reads for the DNA library sample (P1_XbaI_7). For the combined gWT/gMUT libraries, this resulted in approximately 37 million reads combined for the three demultiplexed RNA samples (P1_XbaI_4, P1_XbaI_5, P1_XbaI_6; 9.4–16 million each), and 19.6 million reads for the DNA library sample (P1_XbaI_8). For each library, BCs that had less than three raw counts in any RNA replicate or less than 10 raw counts in the DNA sample were removed before proceeding with downstream analyses.

Expression normalization was performed by first calculating reads per million (RPM) per BC for each replicate for both the SYN library and the combined gWT/gMUT library. For each BC, expression was calculated by dividing the RPMs in each RNA replicate by the DNA pool RPMs for that BC. Normalizing by DNA RPMs successfully removed the impact of the representation of the construct in the original pool as the calculated expression has no correlation with the DNA counts for both the SYN library and the combined gWT/gMUT. Within each biological replicate, the BCs corresponding to each synthetic element (SYN) or genomic sequence (gWT/gMUT) were averaged and then normalized by basal mean expression in that replicate. These normalized expression values were then averaged across biological replicates. All downstream analyses were performed in R version 3.3.3 and plotted with ggplot2 version 2.2.1. Expression summaries per replicate are reported in *Supplementary file 1C* for the SYN library, *Supplementary file 1H* for the gWT/gMUT library and *Supplementary file 1L* for the 'mini spacer' library.

## Predicted occupancy

Custom code, based on Zhao and Stormo's BEEML algorithm (*Zhao et al., 2009*), was used to compare sequences of interest to a provided Energy Weight Matrix (EWM) at a set protein concentration (mu) and output a predicted occupancy for that TF as in *White et al. (2013)*. Briefly, an energy landscape (EWM score) is calculated by comparing all $n$-mers of each sequence, where $n$ = length of provided motif, to the matrix to generate an array of individual base scores for the forward and reverse orientation of the sequence. Occupancy is then predicted using equation 3 for binding probability at equilibrium, $(1/ (1 + e^{(\Delta G - \mu)}))$. Position Frequency Matrices equivalent to the PWMs used for both SYN building block design and for scanning the mouse genome were used to generate EWMs, using the formula $RT * ln( Freq(Base\hat{\ }consensus)/Freq(Base\hat{\ }i))$ to convert the frequency of each base at each position $i$ to a pseudo $\Delta\Delta G$ values for each factor (*White et al., 2013*). Predicted occupancy (P(Occ)) for the 3-mer SYN elements was calculated for different assumed protein concentrations (*mu* = 0.5, 1, 2, 4, 5, 8, 10, 12) to determine at what point the SYN elements are predicted to be saturated, where P(Occ) $\cong$ three for each SYN element, that is: approaching one for each TFBS in the sequence. SYN elements were saturated by each of the four pluripotency factors at *mu* = 8 with the exception of the shorter Oct4 motif, which reached saturation at *mu* = 10. Occupancy of gWT and gMUT sequences was predicted for gWT and gMUT at an assumed high protein concentration of *mu* = 8 for Sox2, Klf4, Esrrb, and *mu* = 10 for Oct4, consistent with the role of these factors in mESCs. The predicted occupancy of each factor for matched gMUT sequences are reported in *Supplementary file 2F* as a feature of gWT sequences. iRF models:

We built iterative Random Forest (iRF) models to classify our data using the R package iRF (version 2.0.0) (*Basu et al., 2018*). To run the software a model is initialized with $1/p$ weights for each of $p$ features to be included in fitting the model. In each iteration, $p$ features are reweighted by their Gini Importance ($w^k$), a measure that is calculated by how purely a node, split by feature, separates the classes (*Menze et al., 2009*; *Louppe et al., 2013*). Default settings were used for model training,

with four iterations of reweighting *p* features specified for each model as indicated in *Supplementary files 2A and 2B*.

Synthetic data was split into training and test sets by randomly subsetting 50% of the total SYN elements (total n = 407). Mean normalized expression was the response variable for model fitting for the synthetic models (see *Supplementary file 2E* for feature annotations for SYN elements). Four iterations of model fitting on training data was used.

Genomic data was split into training and test sets by randomly subsetting 50% of the total gWT/ gMUT intervals (total n = 624). Classification as 'active', 1, if mean normalized gWT expression was greater than or equal to the 3rd quartile and 'inactive', 0, if mean normalized gWT expression was less than the 3rd quartile (cutoff value = 1.983), was the response variable for model fitting (see *Supplementary file 2F* for feature annotations and response values for gWT sequences). Four iterations of model fitting on training data was used. gkm-SVM:

We used a gapped *k*-mer Support Vector Machine (gkm-SVM) to search for gapped k-mers that distinguish between highly active and inactive genomic sequences (*Ghandi et al., 2016*). We subset sequences from the gWT library into top 25% (high) and bottom 25% (low) based on expression data for a total of 101 positive and 101 negative intervals for the training set. FASTA sequences were then generated from the mm10 reference genome (Bioconductor, BioMart) for each region (*Supplementary file 4*). We then used the gkm-SVM R package to classify high vs. low sequences (*Ghandi et al., 2016*). Word length (L) values of 6 (gap = 2), 8 (gap = 2), and 12 (gap = 6), were tested with cross validation. Default settings were used for other function options. Three-fold cross validation was chosen due to the the amount of structure in the data, with combinations of OSK binding sites overrepresented in positive training sequences (*Figure 3—figure supplement 1*). The best average performance on training data as evaluated by AUCs was the model trained with parameters of L = 8 and gap = 2 (See *Supplementary file 2G* for output scores). The final gkmer-SVM model includes approximately 1 million unique *k*-mers (See *Supplementary file 2C* for full kmer list and weights).

## Other analysis and data sources

All genome coordinates from previous mouse genome builds were converted to mm10 using the UCSC liftover tool (*Kuhn et al., 2013*). Binding matrices for SOX2, OCT4, KLF4, ESRRB were as previously reported (*Fiore and Cohen, 2016*). The Bedtools suite (version 2.20) was used for manipulations and analysis of bed files (*Quinlan and Hall, 2010*). Statistical tests were chosen based on expectations of normalcy, with Wilcoxon rank-sum test used for comparisons of BC expression as these distributions were observed to be skewed for some library members, Welch's t-test used where sample sizes were equal and roughly normal, and Fisher's 1-sided tests used for testing for enrichment in small sample sizes.

## Data access

Raw sequencing data for SYN library and gWT/gMUT library can be found under SRA accession number SRR7515851. Processed sequencing data, specifically demultiplexed barcode counts per replicate, can be found under GEO accession number GSE120240. Additionally, a table of normalized reads per million (RPMs) across replicates for all barcodes are included as *Supplementary file 1D* for the SYN library, *Supplementary file 1I* for the gWT/gMUT library, and *Supplementary file 1M* for the MiniSpacer library.

## Acknowledgements

We thank members of the Cohen Lab for critical reading and feedback, particularly Michael White, Max Staller and Hemangi Chaudhari for helpful discussion over the course of the project, Jessica Hoisington-Lopez from the DNA Sequencing Innovation Lab for assistance with high-throughput sequencing, and Karl Kumbier for modeling discussions. This work is supported by a grant from the National Institutes of Health, R01 GM092910 to BAC.

# Additional information

## Funding

| Funder | Grant reference number | Author |
|---|---|---|
| National Institutes of Health | R01 GM092910 | Barak Cohen |

The funders had no role in study design, data collection and interpretation, or the decision to submit the work for publication.

## Author contributions

Dana M King, Conceptualization, Data curation, Formal analysis, Investigation, Visualization, Methodology; Clarice Kit Yee Hong, Data curation, Formal analysis, Methodology, Writing - review and editing; James L Shepherdson, Investigation, Writing - review and editing; David M Granas, Investigation; Brett B Maricque, Data curation, Formal analysis, Methodology; Barak A Cohen, Conceptualization, Supervision, Funding acquisition

## Author ORCIDs

Dana M King (iD) https://orcid.org/0000-0003-4635-5272
Clarice Kit Yee Hong (iD) https://orcid.org/0000-0002-9485-1425
James L Shepherdson (iD) https://orcid.org/0000-0002-3288-7000
Barak A Cohen (iD) https://orcid.org/0000-0002-3350-2715

## Decision letter and Author response

Decision letter https://doi.org/10.7554/eLife.41279.sa1
Author response https://doi.org/10.7554/eLife.41279.sa2

# Additional files

## Supplementary files

- Supplementary file 1. Composition of libraries and expression measurements.
- Supplementary file 2. Features, weights and motifs of iRF and gkmSVM models.
- Supplementary file 3. Sequences of primers used in this study.
- Supplementary file 4. FASTA-format input file of GEN library sequences for gkmSVM.
- Transparent reporting form

## Data availability

Sequencing data has been deposited in GEO under accession code GSE120240. Any additional data generated during this study are included in the manuscript and supporting files.

The following dataset was generated:

| Author(s) | Year | Dataset title | Dataset URL | Database and Identifier |
|---|---|---|---|---|
| King DM, Maricque BB, Cohen BA | 2019 | Massively Parallel Reporter Assay for pluripotency factors in mESCs | https://www.ncbi.nlm.nih.gov/geo/query/acc.cgi?acc=GSE120240 | NCBI Gene Expression Omnibus, GSE120240 |

The following previously published datasets were used:

| Author(s) | Year | Dataset title | Dataset URL | Database and Identifier |
|---|---|---|---|---|
| Chen X, Xu H, Yuan P, Fang F, Huss M, Vega VB, Wong E, Orlov YL, Zhang W, Jiang J, Loh YH, | 2008 | Mapping of transcription factor binding sites in mouse embryonic stem cells | https://www.ncbi.nlm.nih.gov/geo/query/acc.cgi?acc=GSE11431 | NCBI Gene Expression Omnibus, GSE11431 |

| | | | | | |
|---|---|---|---|---|---|
| Yeo HC, Yeo ZX, Narang V, Govindarajan KR, Leong B, Shahab A, Ruan Y, Bourque G, Sung WK, Clarke ND, Wei CL, Ng HH | | | | | |
| Yu HB, Johnson R, Kunarso G, Stanton LW | 2011 | Genome-wide maps of REST and its cofactors in mouse E14 cells | https://www.ncbi.nlm.nih.gov/geo/query/acc.cgi?acc=GSE28233 | NCBI Gene Expression Omnibus, GSE28233 | |
| Perino M, van Mierlo G, Karemaker ID, van Genesen S, Vermeulen M, Marks H, van Heeringen SJ, Veenstra GJC | 2018 | MTF2 recruits Polycomb Repressive Complex 2 by helical shape-selective DNA binding | https://www.ncbi.nlm.nih.gov/geo/query/acc.cgi?acc=GSE94300 | NCBI Gene Expression Omnibus, GSE94300 | |
| Wu J, Huang B, Chen H, Yin Q, Liu Y, Xiang Y, Zhang B, Liu B, Wang Q, Xia W, Li W, Li Y, Ma J, Peng X, Zheng H, Ming J, Zhang W, Zhang J, Tian G, Xu F, Chang Z, Na J, Yang X, Xie W | 2016 | The landscape of accessible chromatin in mammalian pre-implantation embryos (ATAC-Seq) | https://www.ncbi.nlm.nih.gov/geo/query/acc.cgi?acc=GSE66581 | NCBI Gene Expression Omnibus, GSE66581 | |
| Pervouchine DD, Djebali S, Breschi A, Davis CA, Barja PP, Dobin A, Tanzer A, Lagarde J, Zaleski C, See L, Fastuca M, Drenkow J, Wang H, Bussotti G, Pei B, Balasubramanian S, Monlong J, Harmanci A, Gerstein M, Beer MA, Notredame C, Guigo R, Gingeras TR | 2014 | The conserved organization of the human and mouse transcriptomes | https://www.ncbi.nlm.nih.gov/geo/query/acc.cgi?acc=GSE49417 | NCBI Gene Expression Omnibus, GSE49417 | |

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
