## [Decision Letter]

**Acceptance summary:**

We are excited to have this impressive study of *cis*-regulatory grammar published in *eLife*. Figuring out how *cis*-regulatory sequences determine gene expression has been a long-standing challenge for the field, and this study makes an important contribution by revealing the relative roles of transcription factor binding affinity and surrounding sequence context with a rigorous and deep set of experiments.

**Decision letter after peer review:**

Thank you for submitting your article "Synthetic and genomic regulatory elements reveal aspects of *cis*-regulatory grammar in Mouse Embryonic Stem Cells" for consideration by *eLife*. Your article has been reviewed by two peer reviewers, and the evaluation has been overseen by Patricia Wittkopp as both Reviewing and Senior Editor. The following individual involved in review of your submission has agreed to reveal his identity: David N Arnosti (Reviewer #1).

The reviewers have discussed the reviews with one another and the Reviewing Editor has drafted this decision to help you prepare a revised submission.

Summary:

The conversion of DNA sequence information to transcriptional output relies on the context-specific interactions of transcription factors and cofactors, which influence each other and the transcription process in multiple ways. Thus, it has been difficult to identify general principles allowing predictive models of *cis* regulatory element outputs, even with precise information about TF concentrations and DNA sequences acted upon. In this manuscript, King et al. use plasmid-borne massively parallel reporter assays to identify *cis* regulatory considerations that influence the activities of Oct4, Klf4, *Sox2*, and ESRRB transcription factors in embryonic stem cells, which have been well characterized for gene expression and genome-wide chromatin features. They take a two-pronged approach for this study, testing hundreds of elements in plasmid libraries. They create synthetic elements carrying binding motifs for the four proteins in varying numbers (2-4 sites), and they select genomic sequences that resemble these elements in that they are known to have some level of protein occupancy, and bear motifs of the OSKE factors. Using high-throughput sequencing, the activities of the libraries are assessed in transfected cells, and relative expression compared to certain mutant constructs in which the motifs are removed.

The presence of a large number of inactive elements is a valuable finding that allows the authors to assess the importance of specific features for activity vs. inactivity.

The authors determined that the activities of the generally active synthetic elements can be predicted by random forest machine learning models, employing the number of bound elements as well as the binding site arrangements. However, the insights gained from these constructs appear not to contain predictive power on the genomic sequences, where a smaller fraction of candidate elements are active. A gapped kmer approach indicates that differentiating the active from inactive genomic sequences involves the identification of additional binding sites. A more nuanced RF modeling approach involving factor spacing, primary sites, and ChIP signals measured on these elements is able to provide a better level of accuracy than any of these elements alone; interestingly, these are factors that were specifically left out of the synthetic library, where spacing is held constant, and motif quality is not varied. Consequently, when using synthetic elements, an enhanceosome model described their data better, while with genomic elements, the billboard model worked better.

Overall, this study points to possible avenues for progress, as well as very specific reasons for pessimism. The synthetic elements tested include certain features that may not apply to endogenous elements (placement of an element directly next to a basal promoter, plasmid rather than integrated location), as well as consciously avoid variables that may be key (differences in spacing, affinities). Since we already have much evidence for the roles of spacing and motif affinity, it makes sense that the authors deliberately set up a testing situation which can assess other factors for possible use in wider modeling efforts, namely order of elements on the enhancer, and number of factors present. The answer appears to be that any informative synthetic approaches must incorporate the factors pursued in the analysis of their endogenous data elements. Overall, this study makes an important contribution to identification of pathways that must be pursued to subsequently create deeper understanding of the DNA-to-transcriptional output function.

Essential revisions:

1) Differing enthusiasm for the modeling component was reflected in the reviewers' comments. One thought that the emphasis on the models was an over-reach, especially because the data neither supports one or the other model, nor is there enough of it to make a definitive claim. The other was not bothered by the data not fitting neatly into either model nor what was perceived as an oversimplification of the models. However, even this latter reviewer agreed that the authors should more clearly spell out how their tests do or do not sample the many variables. Revision to the modeling section to make these points more clear to readers is needed.

2) There was also a difference in opinion about how much this work advances the field, with one reviewer pointing out that it doesn't identify new physical principles or factors affecting transcriptional regulation, and the other agreeing but arguing that this work is part of the necessary path that our fields must explore to make real progress on predictive approaches. I agree that the large size of the set of active/inactive endogenous elements characterized is a very important contribution to the field; one that will help us better recognize and understand enhancer sequences. Revision to the text to more explicitly articulate the contribution to the field of this work is needed to address this concern.

Below are specific comments from reviewers that elaborate on the concerns expressed more generally in the two points above:

1) Even though I like the logo-like presentation of the preferred order of the TF on the synthetic promoters, to claim that this result fits an enhanceosome model is a stretch at best. An enhanceosome model requires positioning of every TF in a particular conserved structure. Here there is a certain preference for some positioning. To my understanding the authors only used identical/constant non-binding site sequence in all of the synthetic constructs. How do they know that these sequences do not influence the order? Can the authors choose two variants (one strong and one weak), and introduce 5 different flanking sequences and check whether the ratio in expression between both arrangements is conserved?

2) For the genomic elements the results are not surprising, as we expect to get interference from unknown or cryptic regulatory elements. The analysis they provide in Figure 5 shows a nice correlation between ChIP-seq data and strength of expression supporting the notion that the more elements bind the promoter the higher the expression. From my perspective this result supports neither the billboard nor the enhanceosome model, but rather a more dynamic model where the cumulative occupancy keeps the promoter open for longer supporting a higher expression. The dynamic or "occupancy" model is further also supported by the data in Figure 4, where there is a correlation between higher expression and stronger sites. I would like to see a discussion of a more dynamical model as another option for explaining their data.

3) The claim that optimal spacing leads to better expression is also not supported but the data. First of all – what is optimal spacing? The author don't say. Is it constant for all the TFs used, or different for each TF pair? Do they have data which supports one type of spacing over another? Finally, how do they know that next-nearest neighbor effects do alter "optimal spacing" of nearest neighbors? To answer this question will require another much larger OL, which is clearly outside of the scope. Nevertheless, I would like the authors to clarify their claim.

4) The authors implicitly factor out a number of elements in their synthetic library, either to make the task manageable, or because they suspect certain features are more important. These include the design of having up to just one binding motif for each of the factors, the placement of the factors adjacent to the basal promoter, ensuring that certain proteins will have privileged access to the basal machinery, and the decision to not test spacing. The logic of the paper would be easier to follow if the authors would explain why they made these choices e.g. perhaps certain features are already well enough known.

5) The finding that chromatin accessibility is not at all predictive is quite fascinating – many studies have relied on such data to infer where relevant enhancers are, and in which cell types. The authors should place this finding in context – does it have something to do with their use of plasmid-borne genes, rather than integrated reporters?

6) The RF modeling of genomic sequences with the most complex set of features (58 in all) sorts enhancers into active and inactive elements (if I understood their approach). Would the predictions be different, more informative, if they were attempting to predict relative activity? IF this is a misunderstanding, it would be helpful to clarify.

---

## [Author Response]

Essential revisions:1) Differing enthusiasm for the modeling component was reflected in the reviewers' comments. One thought that the emphasis on the models was an over-reach, especially because the data neither supports one or the other model, nor is there enough of it to make a definitive claim. The other was not bothered by the data not fitting neatly into either model nor what was perceived as an oversimplification of the models. However, even this latter reviewer agreed that the authors should more clearly spell out how their tests do or do not sample the many variables. Revision to the modeling section to make these points more clear to readers is needed.

In the revised text we removed statements that our data support either the billboard or enhanceosome models because our results do not neatly confirm the predictions of either model. In the revised Introduction we still discuss these two models in order to summarize current thinking in the field [subsection “Regulatory grammar”, first paragraph]. Discussing these models also helps set the stage for the problem our study addresses, which is the extent to which binding sites function independently or through interactions with each other.

We also clarified which variables are and are not analyzed by the models. We provide justifications for the choices we made in the synthetic library in the section [subsection “Rationale and description of enhancer libraries”, second paragraph.

2) There was also a difference in opinion about how much this work advances the field, with one reviewer pointing out that it doesn't identify new physical principles or factors affecting transcriptional regulation, and the other agreeing but arguing that this work is part of the necessary path that our fields must explore to make real progress on predictive approaches. I agree that the large size of the set of active/inactive endogenous elements characterized is a very important contribution to the field; one that will help us better recognize and understand enhancer sequences. Revision to the text to more explicitly articulate the contribution to the field of this work is needed to address this concern.

In the revised text we attempted to more clearly articulate the contribution our study makes towards a better understanding of gene regulation. We attempted to make three points.

1) While it is known that enhancers are sequences that contain collections of transcription factor binding, we cannot distinguish true enhancers from spurious conglomerations of binding sites. The most dramatic manifestation of this problem is our presentation of a large number of inactive sequences that have the same sequence features as the active sequences. Distinguishing the properties of these two groups is a major challenge for the field.

2) While we know that TFs sometimes act independently and other times engage in cooperative interactions, we cannot reliably predict the effects of sequence perturbations to specific binding sites. Our study is an attempt to organize the qualitative principles we know about into a predictive quantitative framework. We were not necessarily looking for new principles of gene regulation. Our rationale is that the principles we already know about will be predictive once they are properly organized into a quantitative framework. Our work is a step towards that framework.

3) Our study shows that in many cases gene expression is predictable from the sequence of regulatory elements. However, it also shows the large extent to which the principles of gene regulation are context dependent. Our study quantifies the extent to which the context in which binding sites reside influence their activities and formalizes the challenge this will entail.

Below are specific comments from reviewers that elaborate on the concerns expressed more generally in the two points above:1) Even though I like the logo-like presentation of the preferred order of the TF on the synthetic promoters, to claim that this result fits an enhanceosome model is a stretch at best. An enhanceosome model requires positioning of every TF in a particular conserved structure. Here there is a certain preference for some positioning.

In the revised text we removed all statements that our data support either the enhanceosome or billboard model since it is true that our data do not clearly rule out either model. The discussion of these models in the Introduction [subsection “Regulatory Grammar”, first paragraph] is meant only as way to bring out current thinking in the field. In the text we have made it more clear that the logos show certain preferences for certain arrangements, but that the results do not rule out any specific model.

To my understanding the authors only used identical/constant non-binding site sequence in all of the synthetic constructs. How do they know that these sequences do not influence the order? Can the authors choose two variants (one strong and one weak), and introduce 5 different flanking sequences and check whether the ratio in expression between both arrangements is conserved?

To address this point we constructed a small library based on six 4-mer synthetic elements in which we systematically tested four new spacer sequences [subsection “Modeling supports a role for TFBS positions in setting expression level for synthetic elements but not for genomic sequences”, last two paragraphs]. The spacer sequences had small effects on expression, with differences ranging from 0.3-25%. However, these small changes were enough to change the rank order of activities of the sequences with each spacer. Our interpretation of this result [Discussion, last paragraph] is that the spacer sequences have small effects on the independent contribution of each transcription factor binding site, which accounts for the overall small effect of spacer sequences, but that different spacer sequences may influence the interactions that occur between binding sites or introduce new interactions with factors that might occupy the spacer sequences themselves. This interpretation is consistent with our observations that the expression of the genomic elements do not correlate well with their corresponding synthetic elements, and supports the hypothesis that sequences other than the binding sites contribute to gene expression.

2) For the genomic elements the results are not surprising, as we expect to get interference from unknown or cryptic regulatory elements. The analysis they provide in Figure 5 shows a nice correlation between ChIP-seq data and strength of expression supporting the notion that the more elements bind the promoter the higher the expression. From my perspective this result supports neither the billboard nor the enhanceosome model, but rather a more dynamic model where the cumulative occupancy keeps the promoter open for longer supporting a higher expression. The dynamic or "occupancy" model is further also supported by the data in Figure 4, where there is a correlation between higher expression and stronger sites. I would like to see a discussion of a more dynamical model as another option for explaining their data.

We have revised the text to increase discussion of the occupancy model [Abstract; subsection “Regulatory Grammar”, last paragraph; Discussion, first two paragraphs]. We have also revised the text throughout the manuscript to focus on the distinction between independence and interaction, rather than on the difference between billboard and enhanceosome.

3) The claim that optimal spacing leads to better expression is also not supported but the data. First of all – what is optimal spacing? The author don't say. Is it constant for all the TFs used, or different for each TF pair? Do they have data which supports one type of spacing over another? Finally, how do they know that next-nearest neighbor effects do alter "optimal spacing" of nearest neighbors? To answer this question will require another much larger OL, which is clearly outside of the scope. Nevertheless, I would like the authors to clarify their claim.

We have removed the claim about optimal spacing. Results demonstrating slight preferences for certain spacings between specific sites in genomic sequences are now presented [subsection “Site affinity contributes to the activity of genomic sequences”, last paragraph and Figure 4—figure supplement 2] and those preferences are incorporated into our final Random Forest model [subsection “Contributions from sites for other transcription factors”, last paragraph].

4) The authors implicitly factor out a number of elements in their synthetic library, either to make the task manageable, or because they suspect certain features are more important. These include the design of having up to just one binding motif for each of the factors, the placement of the factors adjacent to the basal promoter, ensuring that certain proteins will have privileged access to the basal machinery, and the decision to not test spacing. The logic of the paper would be easier to follow if the authors would explain why they made these choices e.g. perhaps certain features are already well enough known.

The text now contains a more detailed discussion of how we chose which parameters to test in the synthetic library [subsection “Rationale and description of enhancer libraries”, second paragraph and subsection “MPRA of reporter gene libraries”].

5) The finding that chromatin accessibility is not at all predictive is quite fascinating – many studies have relied on such data to infer where relevant enhancers are, and in which cell types. The authors should place this finding in context – does it have something to do with their use of plasmid-borne genes, rather than integrated reporters?

The text now discusses several possible explanations for why TF occupancy data, but not DNA accessibility data correlates with activity in our assays [Discussion, second paragraph].

6) The RF modeling of genomic sequences with the most complex set of features (58 in all) sorts enhancers into active and inactive elements (if I understood their approach). Would the predictions be different, more informative, if they were attempting to predict relative activity? IF this is a misunderstanding, it would be helpful to clarify.

This is a subtle point which we now attempt to clarify in the text [subsection “Modeling supports a role for TFBS positions in setting expression level for synthetic elements but not for genomic sequences”, fourth paragraph]. Because 2/3 of the genomic sequences are inactive, the largest signal in the genomic sequences comes from the difference between active and inactive sequences. There is very little power to detect the differences in relative activity among the 1/3 of active genomic sequences. For this reason, when analyzing genomic sequences, we restricted ourselves to models that attempt to distinguish between active and inactive sequences.